# Intention to Retire in Employees over 50 Years. What is the Role of Work Ability and Work Life Satisfaction?

**DOI:** 10.3390/ijerph16142500

**Published:** 2019-07-13

**Authors:** Prakash K.C., Jodi Oakman, Clas-Håkan Nygård, Anna Siukola, Kirsi Lumme-Sandt, Pirjo Nikander, Subas Neupane

**Affiliations:** 1Unit of Health Sciences, Faculty of Social Sciences, Tampere University, Arvo Ylpönkatu 34, 33520 Tampere, Finland; 2Gerontology Research Center, Tampere University, FI-33014 Tampere, Finland; 3Centre for Ergonomics and Human Factors, School of Psychology and Public Health, La Trobe University, Melbourne, VIC 3086, Australia; 4Faculty of Social Sciences, Tampere University, FI-33014 Tampere, Finland

**Keywords:** intention to retire, work ability, ageing workers, work wellbeing, psychosocial work exposures

## Abstract

Background: We investigated work ability and trajectories of work life satisfaction (WLS) as predictors of intention to retire (ITR) before the statutory age. Methods: Participants were Finnish postal service employees, who responded to surveys in 2016 and 2018 (n = 1466). Survey measures included ITR, work ability and WLS. Mixture modelling was used to identify trajectories of WLS. A generalized linear model was used to determine the measures of association (Risk Ratios, RR; 95% Confidence Intervals, CI) between exposures (work ability and WLS) and ITR. Results: Approximately 40% of respondents indicated ITR. Four distinct trajectories of WLS were identified: high (33%), moderate (35%), decreasing (23%) and low (9%). Participants with poor work ability (RR 1.79, 95% CI 1.40–2.29) and decreasing WLS (1.29, 1.13–1.46) were more likely to indicate an ITR early compared to the participants with excellent/good work ability and high WLS. Job control mediated the relationship between ITR and work ability (9.3%) and WLS (14.7%). Job support also played a similar role (14% and 20.6%). Conclusions: Work ability and WLS are important contributors to the retirement intentions of employees. Ensuring workers have appropriate support and control over their work are mechanisms through which organisations may encourage employees to remain at work for longer.

## 1. Introduction

An ageing population will require extended working lives in comparison to previous generations to ensure an adequate labour supply and financial resources for retirement [1]. Many countries have instigated initiatives to encourage people to delay their retirement, but with mixed success [2]. Retirement choices are complex [3] and influenced by a range of factors (financial incentives to retire early, poor health and working conditions), which require comprehensive exploration to inform strategies to assist with retaining employees [3,4]. Older workers vary significantly in their physical and mental capacities and, as a result, a nuanced approach to retirement age may better ensure that participation rates remain high for a broad range of employees in different occupations. Meanwhile, however, a broader understanding is needed to inform organizations about the influences they have on their employee’s retirement intentions [5]. 

Job satisfaction is a complex issue, and the particular aspects of work that influence an individual’s overall satisfaction vary. Furthermore, the relative importance of different aspects that influence job satisfaction have been demonstrated to vary over the life course and have been shown to influence intention to retire (ITR) [6,7]. The key challenge for the debate on ITR, therefore, is to fully understand the dynamics surrounding a person moving toward retirement age and the job satisfaction factors that have been demonstrated to change and influence the decision to stay in employment or retire. Sufficient support at work and high level of work satisfaction have been identified as important factors in decisions relating to retirement [8]. Similarly, using longitudinal data, von Bonsdorff and colleagues (2010) reported that employees with lower work life satisfaction were more likely to indicate an earlier intention to retire [9]. A Dutch study found higher levels of work engagement as associated with a delayed ITR among employees of an older age group [10]. 

A further predictor of ITR early is poor work ability [7,9,11,12,13]. Work ability concerns the capacity to manage job demands in relation to physical and psychological resources. Work ability at mid-life has been found to predict early retirement due to disability, supporting the need for a life-course approach to sustainable employment. A number of longitudinal analyses of populations have consistently reported that poor work ability is linked with higher risk of early retirement in comparison to those with good or excellent work ability [11,12,13].

Working conditions are an important contributor to the decision-making process on the timing of retirement [9,14,15,16]. For some workers in physically demanding work, an extended working life is challenging, and early exits are common occurrences [9,14], often ending with a disability pension [17]. In addition to the physical environment, psychosocial working conditions have also been identified as an important influence on whether an employee will choose to stay or leave work before the mandated retirement age [18,19,20]. A study of Finnish social and health care employees by Elovainio and colleagues (2007) reported that job demand and job control was correlated to early retirement thoughts [21]. Likewise, a study among Danish employees aged ≥50 years reported that lack of possibilities for development at work was an important factor to induce early retirement thought [19]. Similarly, low job control predicted exiting paid employment among employees aged 50–63 years in 11 European countries [22]. Furthermore, the support at work offers a way to manage work ability among employees and a platform for discussion and planning of their workload. High levels of support at work has been reported to be associated with considerations about retirement intentions [8].

The aim of encouraging working beyond retirement has been of interest to many industrialized nations. The challenge at a workplace is to prevent early retirement among workers and ensure participation until statutory retirement age, which requires further insights into the role of organizations in retaining workers for longer. Work-related factors and work ability are important contributors of ITR before statutory age, however the role of these factors and possible mediating effects have not yet been explored. To contribute to this important area, the current study aimed to investigate work ability and work life satisfaction as predictors of ITR among the employees aged over 50 years. In addition, the potential mediators of the association between psychosocial exposures and ITR were investigated.

## 2. Materials and Methods

### 2.1. Participants and Design

The data for this study were derived from *“Towards a Two-Speed Finland Survey (2tS)”*, collected from employees of the Finnish postal service, which is a large national public sector company with approximately 20,000 employees. The baseline 2tS was conducted in 2016 (n = 2096, 44% response rate) among all the workers aged ≥50 years who did not explicitly deny receiving a call to participate in a survey. The follow up survey was completed in 2018, with a 70% response rate from baseline respondents, n = 1466. We used a follow up survey in the present study. The ethical approval for the study was provided by the Academic Ethics Committee of Tampere Region (Tampere University, approval number: 32/2016).

### 2.2. Measures

#### 2.2.1. Intention to Retire (ITR)

Intention to retire (ITR) indicates intention to retire before the statutory retirement age. The statutory retirement age in Finland is 65 years. ITR was measured through two questions. The first “Will you be able to work until statutory retirement age?” with a five point Likert scale: *“totally true”, “somewhat true”, “not exactly true”, “not true at all”* and *“cannot say”*. Responses were dichotomized into yes (“*totally true”* and *“about true”*) and no (*“not exactly true”* and *“not true at all”)*. The second question “Have you thought you might retire due to health or other reasons before statutory retirement age?” was measured on a four point Likert scale: *“not thought”, “thought sometimes”, “thought often”, “filed application already”*. Responses were similarly dichotomized into no (“not thought”) and yes (“thought sometimes”, “thought often” and “filed application already”). Dichotomized responses were matched and used as a single item with “yes” (responding “yes” in both questions) and “no” response. The development of this variable was adapted from previous research [23,24]. 

#### 2.2.2. Satisfaction in Working Life (WLS)

The satisfaction with working life (WLS) was assessed retrospectively for the following time periods: “15–29”, “30–39”, “40–49”, “50–59” and “60–69” years of age. Responses were collected on a scale from 0 (very dissatisfied) to 10 (very satisfied). As most respondents were 50–59 years of age, the question regarding WLS during “60–69” years of age was excluded. Responses on WLS at “15–29”, “30–39”, “40–49”, “50–59” years were used to detect the trajectories of WLS. The measure of WLS using a single item has been previously validated [25].

#### 2.2.3. Work Ability

Work ability was assessed using a single item [11]. Respondents were asked to rate their current work ability compared to their life’s best using a scale of 0–10, where “0” indicated the worst and “10” indicated the best work ability [26]. The responses were categorized into poor (0–5), reasonable (6–7), good (8–9) and excellent (10). The good and excellent categories were merged and considered as excellent/good work ability. 

#### 2.2.4. Job Support and Job Control

Job support (5 items) was assessed through questions on horizontal (colleagues) and vertical (supervisors) support, and each item was measured on a scale of “0 (low support) to 10 (high support)” (Cronbach’s α = 0.78). The overall score 8–50 was dichotomized into “high” and “low” using the median value (36.0). Job control (4 items) was assessed with questions related to the respondent’s possibility to learn new knowledge and skills (“0”, low—“10”, high), possibility to influence work and working conditions (“0”, never—“3”, usually), experience of doing important and significant work (“0”, never—“5” daily) and having sufficient education to complete the job (“0”, low—“10”, high) (Cronbach’s α = 0.73). The overall score 0–27 was dichotomized into “high” and “low” using the median value (17.0) (adapted from von Bonsdorff et al., 2012) [27].

#### 2.2.5. Other Covariates

Demographic information was collected at baseline. The mean age of participants was 58.4 ± 3.4 and 60% were men. Two occupational categories were used: white- and blue-collar. Working time was assessed as working hours per week. The presence of any physician-diagnosed disease was measured with a yes/no response. Perceived health was assessed using a single item “how do you rate your current health compared to life’s best?” on a scale of “0–10”. Responses were categorized as *good* (9–10), *moderate* (7–8) and *poor/fair* (0–6). Work stress was assessed using a single item “how do you rate the level of your work-related stress?” on a scale of “0–10”. Responses were categorized as low (0–6), moderate (7–8) and high (9–10). 

### 2.3. Statistical Analysis

Mixture modeling (MM) was used to identify the developmental pathways of work life satisfaction (WLS). Latent class analysis (LCA) was used with continuous latent class indicators and user specified starting values based on the continuous responses of WLS at four lifetime points. LCA is a method that identifies within the data the multiple latent classes with a similar development over time [28]. The MM was fitted with two to four classes and the best-fitted model was selected, based on Bayesian Information Criterion (BIC) and Akaike Information Criterion (AIC), substantive interpretability of classes, parsimony and entropy [28,29]. The fit indices are presented in Table 1. The four-class model was selected as it had a lower BIC and lower AIC value and distinct development patterns of all four classes. The four-class model resulted in latent classes that represent low, decreasing, moderate and high WLS, respectively. 

The differences between work ability categories, trajectories of WLS, work and behavior related explanatory factors were examined using χ^2^ test (*p* < 0.05) and analysis of variance. A generalized linear model was used to calculate the Risk Ratio (RR) and their 95% Confidence Intervals (CIs) for the association between exposures and intention to retire. We observed no interaction between gender, age and exposures associated with the outcome, so models were adjusted for age and gender. The final model was adjusted for age, gender, working hours per week, job support, job control, work stress, perceived health status and occupational class. The respective RR estimates were used to calculate the percentage of excess risk mediated (PERM). The selection of confounders and method of calculation of PERM and proportion of risk mediated was adapted from [12]. We estimated the PERM as follows: $$PERM=\frac{RR\;\left(age\;and\;gender\;adjusted\right)-RR\left(fully\;adjusted\right)}{\left[RR\;\left(age\;and\;gender\;adjusted\right)-1\right]}\times \;100.$$

The variables with higher (PERM) values were selected (traditional difference method) [30] and treated as mediators in the generalized structure equation modeling (GSEM). A variable should be representative of a process in a causal chain between the exposure (work ability and trajectories of WLS) and the intended outcome (ITR) to be considered a mediator, which requires the variable to be correlated with both exposure and outcome [31]. GSEM was used to calculate the natural direct and natural indirect effects, and based on those effects, the proportion mediated was calculated for the mediators (often called a counterfactual method). In this analysis, both exposures were constructed and used as binary variables (Work ability: Poor + Moderate versus Good + Excellent & WLS: Low + Decreasing versus Moderate + High). In the traditional analysis, the age and gender adjusted model was treated as a crude model. The final model was controlled for mediators in order to calculate the proportion mediated. In the counterfactual analysis, the models were similarly controlled for age and gender and proportion mediated (by allowing mediators in the model) was calculated using natural direct, natural indirect and total effects. MM was executed in Mplus version 7.11 (Muthen & Muthen, 3463 Stoner Ave., Los Angeles, CA, USA) and all other analyses were executed in STATA 14.0 (StataCorp LP, College Station, TX, USA). 

## 3. Results

Four distinct pathways of WLS were identified: high (33%), moderate (35%), decreasing (23%) and low (9%) (Figure 1). WLS was mostly increasing or constant from 15–29 years to 30–39 years for the low, moderate and high trajectory group. However, for the decreasing trajectory group, WLS was already decreased from 15–29 years with a sharp decline after 30–39 years of age. 

Participant demographics and other work-related characteristics in relation to work ability and trajectories of WLS are described in Table 2. Approximately 40% of the respondents indicated that they intended to retire early, that is before the official retirement age. Excellent/good work ability was reported by 40% of the respondents, followed by moderate (34%) and poor (26%). The age of the respondents was significantly different among trajectory groups. Perceived health of the respondents was significantly different between work ability categories and WLS trajectories. Similarly, responses on job support and job control were significantly different among work ability categories (55% of the respondents with high job support had excellent work ability) and WLS trajectories (42% of the respondents with high job support had high WLS). Likewise, working hours per week and work stress differed according to the work ability and WLS pathway in which an individual belonged. Respondents with good/excellent work ability and high WLS reported lower work stress compared to those with poor work ability and low WLS. 

The estimates for the association (Risk Ratio, RR; 95% Confidence Interval, CI; Percentage of excess risk mediated, PERM) between work ability and ITR with simultaneous adjustments for various characteristics of the study population are described in Table 3. Following adjustments for age and gender, participants with moderate work ability (RR 2.07, 95% CI 1.72–2.51) and poor work ability (RR 3.73, 95% CI 3.14–4.42) had an increased likelihood of indicating ITR early compared to the participants with excellent/good work ability. Following adjustment for age, gender, working hours per week, job support, job control, work stress, perceived health status and occupational class, the estimates were attenuated (RR 1.36, 95% CI 1.09–1.70 for moderate; RR 1.79, 95% CI 1.40–2.29 for poor). Among work-related variables, in a step wise adjustment of the age and gender adjusted model, the adjustment for job support (PERM 11.2% for moderate and PERM 15.0% for poor) and job control (8.4% and 9.9%) contributed to the higher attenuation of these associations. 

The association between trajectories of WLS and ITR are shown in Table 4. Following adjustments for age and gender, participants with moderate WLS had almost similar probability (RR 1.09, 95% CI 0.92–1.29), while those with decreasing (RR 2.26, 95% CI 1.95–2.60) and those with low (RR 1.59, 95% CI 1.30–1.95) had an increased likelihood of indicating ITR early compared to those with a high level of WLS. Following adjustment for age, gender, working hours per week, job support, job control, work stress, perceived health status and occupational class, only those with decreasing WLS had significantly higher probability (RR 1.29, 95% CI 1.13–1.46) compared to those with high levels of WLS. The large proportion of the respondents falling under decreasing WLS had poor perceived health, low job support, low job control, high work stress and were blue collar workers. In addition, decreasing WLS represented almost 50% of respondents with intentions to retire early. Therefore, the decreasing WLS group had a higher risk of early retirement. Among work-related variables, in a step wise adjustment of age and gender adjusted model, the adjustment for job support (PERM 34% for decreasing WLS and PERM 36.0% for Low WLS) and job control (27% and 32%) contributed to the higher attenuation of these associations. 

The largest attenuation of the associations (PERM) presented in Table 3 and Table 4 in stepwise adjustments were due to job support and job control. As a result, these were then checked as mediators in the association between exposures (work ability and WLS) and ITR. In addition, both job support and job control were associated with both exposures and outcome, and therefore considered potential mediators. RR and 95% CI for the association between exposures (work ability and trajectories of WLS) and ITR with job support as mediator and proportions mediated is presented in Table 5. The risk of ITR early was decreased by 14% among those with poor work ability (poor + moderate versus good + excellent) and by 20.6% among those with low WLS (low + decreasing versus high + moderate) when controlled for job support in the model without exposure mediator interaction. Job control as mediator and proportions mediated is presented in Table 6. The risk of ITR early was decreased by 9.3% among those with poor work ability and by 14.7% among those with low WLS when job control was controlled for. The exposure mediator interaction did not describe the association. 

## 4. Discussion

The current study aimed to investigate the role of work ability and trajectories of WLS in predicting ITR amongst employees aged over 50 years of age. Four distinct pathways of WLS were identified: high, moderate, decreasing and low. Most employees were in the high and moderate pathways of WLS and reported excellent/good work ability. However, in relation to the key influences on ITR early, those with poor work ability and decreasing WLS were more likely to indicate an ITR before the pensionable age. However, high levels of support and control at the workplace were found to ameliorate the risk of early retirement.

The comparison with the existing literature suggests that our findings are plausible. In line with previous research, those with lower job satisfaction and poor work ability were more likely to indicate an intention to retire early [7,11,12,13,32]. More specifically, Oakman and Wells (2016) found that the relationship between job satisfaction and ITR early was mediated by work ability, offering organizations an opportunity to design work to enable those with lower work ability to remain at work. An 11-year longitudinal analysis of Finnish employees reported work ability as a stable predictor of ITR early (Von Bonsdorff et al., 2010) [9]. In addition, our study is in line with previous studies in terms of a positive association between high WLS and delayed ITR [5,6,9]. There was a direct transition to early retirement among the people with declining work ability in the U.S. [13]. However, the role of other work-related conditions were not reported in the study. 

The important role of working conditions in reducing the likelihood that employees will retire early was identified in the current study. They are of particular significance and offer insights into potential mechanisms for developing strategies to encourage retention. It is not surprising that job control was also relevant in its association with intention to retire. Allowing individuals to plan their work may enable them to manage some level of incapacity or lower work ability through structuring of their work tasks in a way that they are able to complete. Those with high psychological demands were likely to have ITR early among the participants of a recent Maastricht cohort study [24]. However, they presented an interplay of numbers of other work-related factors and personal factors in taking retirement decisions. The psychological demand used by them is comparable with that of our study, however they are not exactly similar. Lack of job control predicted exit from paid employment among the people aged 50–63 years in 11 countries around Europe [22]. However, the retirement intentions were not reported for those who exited paid employment. Low autonomy in the job was a pushing factor to ITR as early as possible among Norwegian employees aged 60–67 years [18]. Our study supports the findings by Blekesaune & Solem (2005), with the notion that adverse psychosocial attributes increase ITR early and vice versa. Our study similarly corroborates the findings by Thorsen et al. (2012), in which they reported that a lack of possibilities for development at work was an important factor to induce early retirement thought among Danish employees aged ≥50 years [19]. The similarity could be attributed to the similar age group and similar working conditions. However, our study was not able to replicate the gender difference reported in their findings.

In addition, developing interventions to reduce the decline of work ability is an important part of a comprehensive approach to this complex issue of retirement intentions. The relationship of job support and control in reducing the risk of early retirement is an important finding. The literature on psychosocial working conditions has been mixed—that is some have reported potential to delay retirement through organizational actions [18], whilst others have reported no influence [16,33]. The difference could be explained by the variation in the type of industries studied. Interestingly, the prevalence of ITR was almost the same in our study (41%) and the study by Sejbaek et al. (2013) (50%) [16]. The similarity could be explained by the fact that both of the participants belong to Nordic countries and have mostly similar welfare societies for employees. Job support and job control emerged as robust mediators in the pathway of association between work ability, WLS and ITR. The results indicate a longer intention to work among the workers with high job support from their supervisors and colleagues and equally among those perceiving good control of their work. Job control affords individuals the opportunity to tailor their working conditions to suit their capacities and is in line with previous research on person environment (PE) fit, which proposes that the environment should be modified to suit the needs of the individuals rather than the other way around [7,34]. This notion of PE fit is increasingly important to facilitate extended working lives, with a need to adapt the environment to assist workers with changing capacities as they age and creating sustainable employment opportunities. For job control to be effective, support from employers is critical, hence the finding that support was an influence in decisions around retirement is not surprising and is consistent with [8]. Supporting leadership enables discussion and planning of an individual’s workload to ensure that whilst productivity is maintained, it can be done with input from employees in how that is managed. A multi-faceted approach will be required by organizations to encourage the retention of older workers, which takes into account the work environmental factors and an individual’s work ability. This will require communication with workers to determine what the key influences in their decision making are.

### Strengths and Limitations

A key strength of the study is the use of different time points to investigate working life satisfaction. A further strength is the significant size of the participating organization and the nature of the work undertaken in the Postal Service, which is similar in many countries and therefore offers some generalizability beyond the current study. The use of self-reported responses is a source of potential bias. Recall bias is a potential issue in relation to the question on work life satisfaction. However, the authors believe that the identification and use of four different trajectories of WLS provides some level of control on recall bias as individual classes characterize the analogous responses from the study participants. On the other hand, the observed association could have been overestimated given the short time period between measurement of the exposure and outcome variables. Nonetheless, adjustment for other control variables reduced the likelihood of overestimation and influence of responses on exposure to outcome and vice versa. This was additionally checked by using exposure-mediator interactions. The use of a longitudinal design would provide further additional benefit, particularly if participants were followed into retirement to explore the relevant work characteristics that influenced their retirement.

## 5. Conclusions

Workers with poor work ability and decreased work life satisfaction were more likely to indicate an intention to retire early. The risk of intention to retire early among those with poor work ability and those with poor work life satisfaction was lower among the employees with high job support and high job control. For organizations, the current study offers some important insights into strategies to encourage retention of older employees. Good job design, which enables workers’ input into how they work, is likely to reap benefits for both employees and their employers. This includes enabling high levels of control and support for employees to manage their workload. The likely benefit is improved job satisfaction and higher levels of employee retention. In many cases, the workplace could, in fact, serve as an arena to prevent early retirement intentions among the employees, irrespective of their health-related conditions. A supportive workplace could be a platform for employees to continue working until the official age of retirement and further. The findings of the study could be helpful in designing effective interventions to encourage employees to delay their intended timing for retirement.

## Figures and Tables

**Figure 1 ijerph-16-02500-f001:**
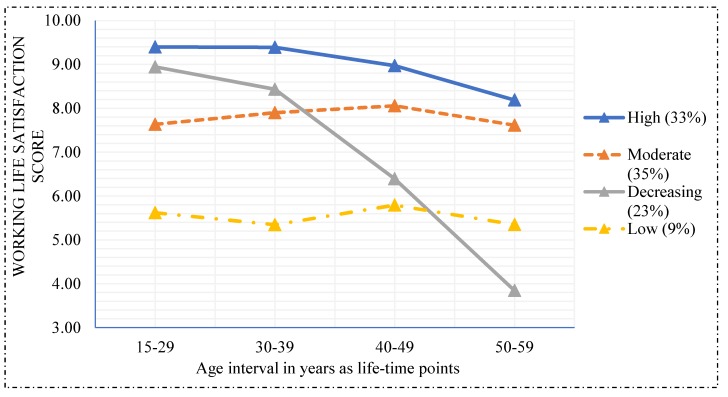
Trajectories of work life satisfaction among the respondents.

**Table 1 ijerph-16-02500-t001:** Fit indices for trajectories of work life satisfaction.

Classes	BIC	AIC	Entropy	Posterior Probability
2	27,466.58	27,371.44	0.73	0.92/0.93
3	26,528.19	26,382.68	0.73	0.91/0.89/0.86
4	25,934.67	25,738.79	0.75	0.87/0.88/0.92/0.84

BIC, Bayesian Information Criteria; AIC, Akaike Information Criteria.

**Table 2 ijerph-16-02500-t002:** Cross tabulation of Basic characteristics of study population and exposure variables.

Characteristics of the Study Population	n = 1466 ^a^	Work Ability (1466) ^b^				Satisfaction in Working Life (n = 1413) ^b,c^				
		Good/Excellent (n = 590) %	Mode-Rate (n = 503) %	Poor (n = 363) %	*p*-Value ^d^	High (n = 466) %	Mode-Rate (n = 498) %	Decreasing (n = 325) %	Low (n = 124) %	*p*-Value ^d^
**Gender**										
Women	587	42	31	27	0.082	35	38	21	6	0.009
Men	879	39	37	24		32	33	24	11	
**Age (years)**										
51–53	327	47	34	19	0.053	31	36	24	9	0.001
54–56	435	40	33	27		32	31	26	11	
57–59	397	36	37	27		31	37	25	7	
≥60	293	40	35	25		39	40	13	8	
**Occupational Class**										
White-Collar	188	63	24	13	<0.001	38	50	6	6	<0.001
Blue-Collar	1264	37	36	27		32	33	26	9	
**Job Support**										
High	668	55	32	13	<0.001	42	42	9	7	<0.001
Low	786	28	37	35		25	29	35	11	
**Job Control**										
High	659	54	30	16	<0.001	43	41	11	6	<0.001
Low	797	30	38	32		25	31	33	11	
**Perceived Health**										
Good	469	82	15	3	<0.001	44	38	10	8	<0.001
Moderate	684	28	55	17		31	40	21	8	
Poor	302	4	18	78		19	21	48	12	
**Working hrs/ week**										
3–35	187	31	28	41	<0.001	27	32	24	16	0.004
36–40	1066	42	36	22		34	36	22	8	
>40	129	43	35	22		29	36	28	7	
**Work stress**										
Low	884	45	32	23	<0.001	37	35	18	10	<0.001
Moderate	432	35	39	26		29	38	27	6	
High	141	27	35	38		19	26	44	11	

Notes: **^a^** Column total is not equal to N in some variables; **^b^** row percentage; **^c^** n = 1413 due to selection for developmental pathways; **^d^** χ^2^-test.

**Table 3 ijerph-16-02500-t003:** Association between work ability and intention to retire with simultaneous adjustments for different characteristics of the study population.

Models	Good/Excellent Work Ability	Moderate Work Ability		PERM	Poor Work Ability		PERM
	RR	RR	95% CI	%	RR	95% CI	%
Adjusted for age + gender	1.0	2.07	1.72–2.51	Reference	3.73	3.14–4.42	Reference
+Occupational class	1.0	2.05	1.69–2.48	1.9	3.65	3.07–4.34	2.9
+Perceived Health	1.0	1.58	1.28–1.95	45.8	2.01	1.56–2.60	63.0
+Job control	1.0	1.98	1.63–2.41	8.4	3.46	2.89–4.14	9.9
+Job support	1.0	1.95	1.61–2.37	11.2	3.32	2.77–3.97	15.0
+Working hours/week	1.0	2.02	1.66–2.45	4.7	3.62	3.05–4.30	4.0
+Work stress	1.0	2.03	1.67–2.45	3.7	3.59	3.02–4.27	5.1
+All above factors	1.0	1.36	1.09–1.70	66.4	1.79	1.40–2.29	71.1

Notes: RR, Relative Risk; CI, Confidence Interval; All separate analyses are adjusted with age and gender; PERM, Percentage of excess risk mediated (age and gender adjusted estimate used as referent group for calculation); WLS, satisfaction in working life.

**Table 4 ijerph-16-02500-t004:** Association between satisfaction in working life (WLS) and intention to retire with simultaneous adjustments for different characteristics of the study population.

Models	High WLS	Moderate WLS		Low WLS		PERM	Decreasing WLS		PERM
	RR	RR	95% CI	RR	95% CI	%	RR	95% CI	%
Adjusted for age + gender	1.0	1.09	0.92–1.29	1.59	1.30–1.95	Reference	2.26	1.95–2.60	Reference
+Occupational class	1.0	1.07	0.89–1.29	1.48	1.17–1.87	18.6	2.10	1.80–2.46	12.7
+Perceived Health	1.0	1.05	0.89–1.23	1.14	0.93–1.39	76.3	1.35	1.17–1.55	72.2
+Job control	1.0	1.03	0.86–1.24	1.40	1.11–1.77	32.2	1.92	1.62–2.28	27.0
+Job support	1.0	1.04	0.86–1.24	1.38	1.09–1.73	35.6	1.83	1.55–2.16	33.9
+Working hours/week	1.0	1.05	0.88–1.27	1.46	1.15–1.86	22.0	2.11	1.80–2.48	11.9
+Work stress	1.0	1.05	0.88–1.26	1.49	1.18–1.88	16.9	2.04	1.73–2.42	17.4
+All above factors	1.0	1.07	0.92–1.26	1.07	0.91–1.26	88.0	1.29	1.13–1.46	77.0

Notes: RR, Relative Risk; CI, Confidence Interval; All separate analyses are adjusted with age and gender; PERM, Percentage of excess risk mediated (age and gender adjusted estimate used as referent group for calculation).

**Table 5 ijerph-16-02500-t005:** Risk Ratio (RR) and 95% CI on the association between exposures (work ability and satisfaction in work life (WLS)) and outcome (intention to retire) with Job support as the mediator.

Method of Analysis	RR	95% CI	Proportion Mediated (%)
Traditional analysis for work ability			
Poor + Moderate versus Good + Excellent work ability ^a^	2.78	2.34–3.30	Reference
Poor + Moderate versus Good + Excellent work ability ^b^	2.50	2.10–2.98	15.7
Counterfactual analysis ^a^			
Good + Excellent versus Poor + Moderate work ability			
(effect), without exposure mediator interaction			
Direct effect	1.37	1.31–1.44	
Indirect effect	1.04	1.03–1.06	
Total effect	1.43	1.36–1.50	14
Good + Excellent versus Poor + Moderate work ability			
(effect), with exposure mediator interaction			
Direct effect	1.30	1.21–1.39	
Indirect effect	1.02	1.00–1.05	
Total effect	1.33	1.23–1.44	9
Traditional analysis for WLS			
Low + Decreasing versus High + Moderate WLS ^a^	1.92	1.71–2.16	Reference
Low + Decreasing versus High + Moderate WLS ^b^	1.68	1.48–1.90	26.1
Counterfactual analysis ^a^			
High + Moderate versus Low + Decreasing WLS (effect),			
without exposure mediator interaction			
Direct effect	1.27	1.11–1.23	
Indirect effect	1.06	1.03–1.08	
Total effect	1.34	1.27–1.41	20.6
High + Moderate versus Low + Decreasing WLS (effect),			
with exposure mediator interaction			
Direct effect	1.19	1.08–1.31	
Indirect effect	1.05	1.02–1.07	
Total effect	1.24	1.12–1.38	21

Notes: RR, Risk Ratio; CI, Confidence Interval; **^a^** Adjusted for age and gender; **^b^** Adjusted for age, gender and job support.

**Table 6 ijerph-16-02500-t006:** Risk Ratio (RR) and 95% CI on the association between exposures (work ability and satisfaction in work life (WLS)) and outcome (intention to retire) with Job Control as the mediator.

Method of Analysis	RR	95% CI	Proportion Mediated (%)
Traditional analysis for work ability			
Poor + Moderate versus Good + Excellent work ability ^a^	2.78	2.34–3.30	Reference
Poor + Moderate versus Good + Excellent work ability ^c^	2.58	2.16–3.08	11.2
Counterfactual analysis ^a^			
Good + Excellent versus Poor + Moderate work ability			
(effect), without exposure mediator interaction			
Direct effect	1.39	1.32–1.46	
Indirect effect	1.03	1.01–1.04	
Total effect	1.43	1.36–1.50	9.3
Good + Excellent versus Poor + Moderate work ability			
(effect), with exposure mediator interaction			
Direct effect	1.36	1.27–1.46	
Indirect effect	1.02	1.00–1.04	
Total effect	1.40	1.29–1.51	10
Traditional analysis for WLS			
Low + Decreasing versus High + Moderate WLS ^a^	1.92	1.71–2.16	Reference
Low + Decreasing versus High + Moderate WLS ^c^	1.74	1.53–1.98	19.6
Counterfactual analysis ^a^			
High + Moderate versus Low + Decreasing WLS (effect),			
without exposure mediator interaction			
Direct effect	1.29	1.22–1.37	
Indirect effect	1.04	1.02–1.06	
Total effect	1.34	1.27–1.42	14.7
High + Moderate versus Low + Decreasing WLS (effect),			
with exposure mediator interaction			
Direct effect	1.29	1.17–1.43	
Indirect effect	1.04	1.01–1.06	
Total effect	1.34	1.21–1.49	14.7

Notes: RR, Risk Ratio; CI, Confidence Interval; **^a^** Adjusted for age and gender; **^c^** Adjusted for age, gender and job control.

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
