# Peer review of "Intention to Retire in Employees over 50 Years. What is the Role of Work Ability and Work Life Satisfaction?"

_ijerph, 2019, doi:10.3390/ijerph16142500_

Round 1

Reviewer 1 Report

The authors investigated the impacts of work ability and work life satisfaction on the intention to early retirement. Specifically, they classified the WLS with different patterns and analyzed the differences in its impacts. However, several aspects should be paid more attention.

Data:

1.     For line 78-81, you mentioned your data source. My understanding (after reading the later part of this paper) is the author only used the survey data for year 2018. If so, is it necessary to mention the data of 2016? The description of 2016 data may make the reader think you are conducting analysis with a longitudinal data.

2.     Why didn’t you use the data in 2016 in order to conduct a longitudinal study?

3.     Since you use the data in 2018 (cross-sectional data), do you think the way you mention causal pathway (line 266) convincing? You also mentioned in line 252-235, that the adverse psychosocial attributes increase ITR early and vice versa, it seems confusing about the causal pathway.

Measures:

1.     For line 95-100, do you have any literature supporting the way you assessing the WLS? WLS is one of your variables of interests, the definition and classification of the 4 patterns is of great importance. At least there is necessity to discuss why you think it is reasonable to identify the 4 distinct pathways in results section.

2.     For line 107-114, the authors described how to construct variables of job support and job control. However, only the dimension of education was supported by prior literature. The reference should be provided in order to improve the rationality of choosing or constructing measurements.

3.     For line 116-122, similar to above question, the reference should be provided in order to strengthen why the author choose these controls (confounding effects).

4.     In the part of statistical analysis, especially line 143-149, the authors just mentioned mediation analysis with GSEM in Stata and related effects. However, a detailed procedure should be provided.

Results:

1.     Figure 1 should be revised. For example, the title of low WLS is not shown.

2.     It needs to explain why decreasing WLS group always have higher probability of early retirement in either results or discussion.

3.     For line 187, the group with decreasing WLS has higher probability of early retirement while the low WLS groups does not show the same pattern just for the model with all controls included. It is very interesting to explain why? Since in table 4 the above models show that even though decreasing WLS group has higher probability, the low WLS group also have high probability of retiring earlier. In other words, the result for last model in table 4 is different from above models and need to be clarified.

4.     Following question 3 in results section, I would suggest you revise the results tabulation in table 4 and exchange the position of low WLS and decreasing WLS. As shown in your figure 1 and paragraph 1 of results section, the decreasing WLS has lowest WLS compared to other 3 groups at their 50s. It would make your results easier understanding and the pattern and differences among different groups would be more clear.

5.   In table 5 & 6, whey the total effect is not the sum of direct effect and indirect effect?

Discussion:

1.     For line 233-234, the author mentioned that work ability mediated the relationship between WLS and ITR, however, this is not illustrated in the analysis or results. Why didn't you follow prior studies? Relationship between job satisfaction and work ability should be examined.

Reviewer 2 Report

This is an interesting manuscript describing a study exploring a novel set of predictors of early retirement intention. The study highlights an important role for psychosocial working conditions in altering the strength of associations between work ability and work life satisfaction on the one hand and intention to retire on the other. In doing so the study flags up the scope for organisations to better manage the psychosocial work environment with a view to reducing intention to retire. 

A minor recommendation: The psychosocial working conditions of control and support are referred to in the keywords as ‘work exposures’. I’d recommend a little more precision here; perhaps use the term ‘psychosocial work exposures’ or ‘psychosocial work characteristics’ in order to make clear that physical work characteristics were not addressed. 

The introduction is clear and concise, draws on the relevant contemporary literature and presents a strong rationale for the selection of the predictor variables. Paragraph beginning line 49: It would be useful to define work ability as it’s possible that not all readers will be familiar with the (relatively new) construct. This could be achieved in relatively few words in the following way: “Work ability (WA) concerns the capacity to manage job demands in relation to physical and psychological resources.” Line 73: Change ‘exposures’ to ‘psychosocial hazard exposures’ or ‘psychosocial work exposures’ in order to make clear that the focus is on psychosocial rather than physical exposures. 

The method is clear and sufficiently detailed to permit replication. The attrition rate between T1 and T2 was low, which helpfully ensured a large sample of participants that completed both questionnaires. Section 2.1 explains that the survey was administered to workers aged 50 and over, with the purpose of identifying predictors of intention to retire prior to statutory retirement age. I’d like to see mention in this section of the actual statutory retirement age; this will help to give the reader an idea of how close (or not) respondents are to that age. 

Line 90: The second item used to assess retirement intention is described as a 5-point scale whereas in fact there are only four response options. 

Line 96: Satisfaction with working life was assessed retrospectively for four periods of working life, going back to age 15. I’d like to see a citation to justify this approach, with a specific focus on the reliability of such retrospective accounts. 

Figure 1: The low trajectory is missing from the key. 

The results and discussion are appropriately presented, with thoughtful consideration given to both the theoretical and practical implications of the study. Care has been taken to avoid overstating the magnitude and implications of the findings.  

The manuscript is beautifully presented and written in an excellent academic style. A joy to review! 
